# Calibrated muscle models improve tracking simulations without enhancing gait predictions

**Filippo Maceratesi**[1,2*], **Míriam Febrer-Nafría**[1,2], **Josep M. Font-Llagunes**[1,2]

**1** Department of Mechanical Engineering and Institute for Research and Innovation in Health (IRIS), Universitat Politècnica de Catalunya - BarcelonaTech (UPC), Barcelona, Spain, **2** Institut de Recerca Sant Joan de Déu, Esplugues de Llobregat, Barcelona, Spain

* filippo.maceratesi@upc.edu

**Data availability statement:** All relevant data are within the manuscript and its Supporting information files.

## Abstract

*Objectives*: This study presents two main aims: (i) to assess functionally-calibrated musculoskeletal models (FCMs) in both tracking and predictive simulations of human motion, against non-linearly scaled models (NSMs), and (ii) to examine the effects of three different variations of our baseline functional calibration approach on the results of tracking and predictive simulations. *Methods*: Motion capture experiments of six functional activities were performed with three healthy subjects. The musculotendon (MT) parameters of 18 muscles per leg were estimated using an optimal control problem. A baseline problem formulation and three variations were developed to generate four different FCMs per subject. Then, the FCMs were compared against NSMs in tracking simulations of the motions excluded from the calibration and fully-predictive simulations of gait. *Results*: In the tracking simulations, the FCMs led to more accurate joint torques estimations. Including gait in the calibration problems improved the knee torques accuracy (normalised root mean square error: $0.31 \pm 0.11$), compared to the baseline calibration (normalised root mean square error: $0.70 \pm 0.21$). Regarding the gait predictive simulations, the NSMs consistently yielded more accurate subtalar inversion/eversion torques and knee flexion angles, compared to the FCMs. The accuracy of the predicted muscle excitations was generally consistent between NSMs and FCMs. *Conclusion*: The results suggest that, while the FCMs led to more accurate joint torques estimations in the tracking simulations, they did not outperform the NSMs in the fully-predictive gait simulations.

## Introduction

Predictive simulations of human motion based on musculoskeletal models can be used to compute the movement mechanics and energetics without relying on experimental movement data. Therefore, this tool might elucidate important human motion principles and the influence of isolated musculoskeletal features, which can assist decision-making processes in clinical settings. The predictions are typically based on the assumption that the central nervous system optimizes the movement with some criteria. For this reason, optimization algorithms are usually employed to predict human motion. Different optimization techniques

**Funding:** Agencia Estatal de Investigación (MCIN/AEI/10.13039/501100011033) and the European Regional Development Fund (ERDF "A way of making Europe"), (Grant PID2021-123657OB-C33), to J.M.F.; AGAUR-FI ajuts, (2022 FI-B 00911), to F.M.

**Competing interests:** The authors have declared that no competing interests exist.

have been used in literature to perform these predictive simulations: reinforcement learning (e.g., [1,2]), direct shooting (e.g., [3]), multiple shooting (e.g., [4,5]), direct collocation (e.g., [6,7]), etc. Direct collocation methods enable efficient and numerically stable solutions of predictive simulations by formulating the dynamics, controls, and states as a large but sparse nonlinear programming problem that can be solved using gradient-based optimization [8]. Musculoskeletal models are computational representations that simulate the interaction between muscles, bones, and external forces. They describe how neural signals activate muscles to generate forces, how these forces are transmitted through the musculoskeletal system to produce movement, and how the skeletal system responds dynamically to both muscle-driven and external forces. The employment of subject-specific models in gait predictive simulations could constitute a promising tool for selecting, adapting and personalizing rehabilitation procedures. However, the parameters of the muscle models typically represent the muscles of an average human, rather than a specific individual [8]. In order to effectively use these simulations in clinical applications, it is crucial to determine subject-specific muscle parameters that accurately represent an individual's muscular properties through non-invasive approaches.

The most-widely used approach to represent musculotendon (MT) actuators in human motion simulations is the Hill-type model [9]. It defines the force generating capacity of a muscle, based on five parameters: the maximal isometric muscle force ($F_{max}^M$), the optimal muscle fiber length ($l_o^M$), the tendon slack length ($l_s^T$), the maximal muscle fiber velocity ($v_{max}^M$), and the optimal pennation angle ($\alpha_o$). According to [10–14], the muscle force estimation is mostly sensitive to the $l_o^M$ and $l_s^T$ values and, therefore, these are considered the most important parameters to personalize (or calibrate) for generating a subject-specific musculoskeletal model. Two overarching methods are employed in literature to personalize muscle models: anthropometric and functional approaches. In the former method, the MT parameters are scaled based on the skeletal dimensions of the specific subject. The simplest anthropometric approach consists in linearly scaling these parameters using software packages such as OpenSim [15]. However, other scaling approaches that preserve the operating conditions of the muscles were proven more accurate [16]. Modenese et al. [17] generalized the previous approach to calibrate the parameters for a full 3D lower limb model. However, since the MT properties vary with age, gender, and activity level, the subject-specific parameters cannot be accurately determined based on anthropometric dimensions only [18,19]. For this reason, functional methods could potentially lead to a more accurate estimation of MT parameters, since they rely on additional experimental measurements.

Functional approaches consist in optimizing the MT parameters, while minimizing the difference between experimental and model-based joint torques. Functional procedures can be based on either maximal or submaximal muscle contractions, which rely on experimental measurements during isometric/isokinetic dynamometry tests and daily activities, respectively. Hatze [20] was the first researcher to estimate MT parameters using maximum isometric tests of the triceps muscle group. Then, Garner and Pandy [21] expanded this method by employing an optimisation algorithm to determine the parameters of 42 muscles actuating 13 degrees of freedom (DOFs) in the upper body. Wu et al. [22] used this method to estimate subject-specific parameters and compared the resulting shoulder muscle forces and glenohumeral joint loading to those generated by generic and scaled-generic musculoskeletal models. Heinen et al. [23] further extended the algorithm of Garner and Pandy [21] to determine subject-specific MT parameters for the lower extremity, based on both isometric and isokinetic tests. Moreover, other studies used the functional method based on submaximal muscle contractions. The first work that employed this approach was performed by Lloyd et al. [24], who personalized 13 muscles of a 1-DOF EMG-driven model of the knee, relying

both on dynamometry and daily activities. Sartori et al. [25] improved the previous approach by investigating the impact of calibrating EMG-driven models with 1 or 4 DOFs on the estimation of joint torques in one leg. Falisse et al. [26] enhanced the optimisation approach formulated by Lloyd et al. [24] by using the direct collocation technique to increase the efficiency of the problem. In addition, they identified the most appropriate set of motions needed to accurately personalize the MT parameters of 8 healthy subjects.

Regarding the evaluation of these calibration methods, two main strategies are adopted in literature. The first one relies on comparing the estimated MT parameters to cadaveric measurements or published reference values [17,21,27]. The other evaluation approach consists in analyzing the model-generated joint torques against the reference quantities calculated through inverse dynamics (in daily activities) [24–26,28–30], or dynamometer measurements (in isometric/isokinetic tests) [23,24]. In the case of daily activities, these model-generated torques are computed through tracking simulations of the analyzed motion. Since this is a restrictive problem, where the only estimated quantities are the joint torques, the sole purpose of these evaluations is to assess whether the calibrated model can achieve accurate torques.

The studies that calibrate the MT parameters only based on maximal muscle contraction activities have two major limitations. Firstly, it is difficult to obtain a complete set of dynamometry measurements, without tiring the subject and causing muscle fatigue. Secondly, according to Wesseling et al. [31], maximal muscle contractions are never reached in practice. On the other hand, the studies developing and testing submaximal muscle contraction-based procedures have a few differences in their approaches. Firstly, the bounds of the solution space of the MT parameters vary significantly between each study. For instance, in [32], they allow the parameters to vary between 50% and 200% of their generic values; whereas, in [25], the parameters are constrained within $\pm2.5\%$ (for $l_o^M$) and $\pm5\%$ (for $l_s^{lT}$) of their initial values. Secondly, the functional activities used in the calibration problems vary significantly between studies. Some only measure walking and static trials (e.g., [28,29,33]), while others use a more vast set of motions [25,26]. Notably, the results in [26] suggest that gait is an essential motion to include in the calibration problems. Thirdly, the number of DOFs in the legs considered in the muscle personalization procedures varies considerably between studies. For example, in [34], they minimize the difference between the experimental and model-based joint torques in only three DOFs. Meyer et al. [28], on the other hand, consider 5 DOFs per leg. Taking into account the torques in different anatomical planes and DOFs might be an important factor in the calibration problems to ensure an accurate estimation of subject-specific MT parameters. All these differences in approaches are yet to be explored. Furthermore, although evaluating the accuracy of the joint torques estimations of the calibrated models in tracking simulations is important for validation purposes, it is crucial to assess these models in fully predictive simulations. In fact, this allows us to examine how the muscle calibration method influences the simulations that do not rely on experimental data. It is important to verify whether the optimal solution found by the predictive problem changes significantly when using different sets of personalized MT parameters. Thus, we can examine if the muscle calibration method affects the solution of the gait prediction, independently of the optimality criteria and constraints set in the problem. Table 1 summarizes the calibration approach, estimated MT parameters, subjects, model and evaluation strategy of a selection of studies that develop and assess a method to personalize MT parameters. An extended version of this table including more studies and information is provided in the supplementary material (S1 Table).

Based on the aforementioned considerations, this study presents two main aims. Firstly, we assess the effects of calibrating MT parameters using a functional method in both tracking

**Table 1. Summary of the previous studies that develop and evaluate a method to calibrate MT parameters.**

| Type of approach | Ref. | Personalized MT parameters | Subjects | Calibrated muscles | Evaluation |
|---|---|---|---|---|---|
| Anthropometric | Manal [27] | $l_s^T$ | 1 healthy | 1 muscle (brachioradialis) | Model parameters vs. reference values [35] |
| | Winby [16] | $l_o^M$, $l_s^T$ | 10 healthy | 13 knee muscles | / |
| | Modenese [17] | $l_o^M$, $l_s^T$ | 2 healthy | (i) 46 leg muscles; (ii) 19 hip muscles | Model parameters vs. cadaveric measurements |
| Functional (maximal) | Hatze [20] | $F_{max}^M$, $l_o^M$, others | 3 healthy | 3 triceps muscles | / |
| | Garner [21] | $F_{max}^M$, $l_o^M$, $l_s^T$ | 3 healthy | 42 arm muscles | Model parameters vs. anatomical study |
| | Van Campen [36] | $l_o^M$, $l_s^T$ | 1 "virtual" | 13 knee muscles | / |
| | Wu [22] | $F_{max}^M$, $l_o^M$, $l_s^T$, $v_{max}^M$ | 6 healthy | 26 shoulder muscles | / |
| | Heinen [23] | $F_{max}^M$, $l_o^M$, $l_s^T$ | 1 healthy | 25 leg muscles | Model torques vs. experimental torques |
| Functional (maximal and submaximal) | Lloyd [24] | $F_{max}^M$, $l_s^T$ | 6 healthy | 13 knee muscles | Model torques vs. experimental torques |
| Functional (submaximal) | Sartori [25] | $F_{max}^M$, $l_o^M$, $l_s^T$ | 1 healthy | 32 leg muscles | Model torques vs. experimental torques |
| | Pizzolato [34] | $F_{max}^M$, $l_o^M$, $l_s^T$ | 5 healthy | 32 leg muscles | Model torques vs. experimental torques |
| | Falisse [26] | $l_o^M$, $l_s^T$ | 8 healthy | 12 knee muscles | Model torques vs. experimental torques |
| | Meyer [28] | $l_o^M$, $l_s^T$, EMG scaling factors, activation time constants, electromechanical delays, activation non-linearity constants | 1 post-stroke | 35 leg muscles | Model torques vs. experimental torques |
| | Falisse [32] | $l_o^M$, $l_s^T$ | 1 child with cerebral palsy | 43 leg muscles | Gait predictive simulation results vs. experimental kinematics, joint torques, muscle activity, and stride lengths |
| | Ao [29] | $l_o^M$, $l_s^T$, EMG scaling factors, activation time constants, electromechanical delays, activation non-linearity constants | 2 post-stroke | (i) 34 leg muscles; (ii) 33 leg muscles | Model torques vs. experimental torques |
| | Akhundov [30] | $F_{max}^M$, $l_o^M$, $l_s^T$, $\alpha_o$, activation and de-activation time constants | 9 healthy | 34 leg muscles | Model torques vs. experimental torques |

and predictive simulations, and compare these results to those obtained using an anthropometric method, for three subjects. The tracking simulations are used to evaluate the models' capability of estimating joint torques (similarly to the aforementioned studies). Conversely, predictive gait simulations are used to assess the models by also examining the predicted kinematics and muscle excitations. We examine the differences between the two types of models and investigate whether these discrepancies between them vary when using tracking or predictive simulations. Secondly, three variations of our baseline muscle personalization method are developed to explore the effects of: (i) allowing more freedom to the MT parameters in the solution space, (ii) including gait in the personalization problem, and (iii) reducing the number of DOFs in the minimization between the experimental and models' torques.

## Materials and methods

### Experimental measurements and data processing

The experimental protocol was approved by the Ethical Committee of the Universitat Politècnica de Catalunya (approval date: March 22, 2023; ID code: 2023.03). The recruitment of

the volunteers began on March 23, 2023 and ended on June 1, 2023. Three healthy subjects gave written informed consent to participate in the experiments, which were divided in two parts: (i) maximum isometric tests, and (ii) motion capture measurements. In the first part, an electromyography (EMG) device (Biometrics Ltd. PS1800, Cwmfelinfach, Wales) was used to measure the electrical activity of 8 muscles per leg (soleus, tibialis anterior, gastrocnemius medialis, vastus lateralis, rectus femoris, biceps femoris, semitendinosus and gluteus medius). The electrodes were placed following the guidelines of Surface ElectroMyoGraphy for the Non-Invasive Assessment of Muscles (SENIAM) [37]. A handheld dynamometer (MicroFET 2, Hoggan Scientific, Salt Lake City, UT, USA) was used to record the joint torques during maximum isometric tests of the knee flexors, knee extensors, ankle dorsiflexors, ankle plantar flexors, and hip abductors. The measurements were taken at three different angles per joint. Furthermore, for the second part of the experiments, the subjects were instrumented with 50 reflective markers, along with the EMG electrodes previously placed. A motion capture system (OptiTrack V100:R2, NaturalPoint Inc., Corvallis, OR, USA) and ground-embedded force plates (AccuGait, AMTI, Watertown, MA, USA) were used to record the markers' trajectories and the reaction loads on the feet, respectively. Six functional activities were carried out by the participants: squat, stair descent, stair ascent, sit-to-stand-to-sit, squat jump and gait. The motions were selected based on a similar study [26], because they agreed with the following criteria: they encompassed a wide range of contractile conditions, required large joint torques, required a large range of joint angles, reflected various muscle-tendon force distributions, and were easily achievable in practice. The dataset of five motions was used to calibrate the MT parameters, while the recordings of the sixth motion, excluded from the calibration, were intended for evaluation purposes only. In addition, a static trial was recorded to measure the weight of the individual and for scaling the musculoskeletal model.

Following the experimental procedures, the raw measurements were processed to allow them to be input into MATLAB and OpenSim. Firstly, the marker trajectories were interpolated when the recordings contained some gaps. Secondly, the ground reaction forces were filtered using a 4th-order zero phase lag Butterworth filter with cut-off frequency of 6 Hz, in agreement with several papers (e.g., [25,28]). Furthermore, EMG signals were also processed following the procedures described in [26]. Firstly, they were band-bass filtered at 20–400 Hz. Secondly, a full-wave rectification was carried out to convert all negative amplitudes into positive amplitudes. Next, low-pass filtering (10 Hz) was applied using a 2nd-order Butterworth filter. Lastly, the EMG data of each muscle were normalized to the peak value achieved during the whole duration of the experiments. A further normalization was performed during the calibration of the MT parameters (see the subsection regarding the muscle calibration problem formulation), because true maximal contractions are extremely difficult to achieve in practice. Since some muscles shared the same innervation and contributed to the same actuation, they were assumed to have the same EMG pattern. Consequently, the 16 measured EMG profiles were attributed to 36 MT units in the model (see S1 Fig in S1 File).

## Musculoskeletal modeling and OpenSim analyses

A generic OpenSim model suitable for high hip flexion tasks (e.g., squatting) [38] was employed for this study. The model included 23 DOFs and 40 Hill-type muscle models per leg. The head, arms and trunk were considered as a single segment. The musculoskeletal system was scaled to match the anthropometric dimensions of the three subjects using the 'Scale Tool' in OpenSim. This process linearly scaled the $l_o^M$ and $l_s^T$ values of each muscle based on the change in MT lengths after scaling. Moreover, inverse kinematics (IK) and inverse dynamics (ID) analyses were carried out on OpenSim to compute the reference data

for the calibration problems and model evaluations. The marker trajectories were input into the IK tool to obtain the MT lengths, moment arms, generalized coordinates, velocities and accelerations. On the other hand, the joint torques and residual loads were calculated through ID.

The muscle activation dynamics reflected the relationship between muscle excitation $e$ and activation $a$ with a first-order differential equation [9]:

$$\frac{da}{dt} = \frac{1}{\tau_a}e - \frac{1}{\tau_d}a + (\frac{1}{\tau_d} - \frac{1}{\tau_a})ea \tag{1}$$

where, $\tau_a = 15$ ms and $\tau_d = 60$ ms are the activation and deactivation time constants, respectively. Furthermore, as mentioned above, the contraction dynamics was defined by Hill-type models, characterized by active force-length ($\mathbf{f}^l$), passive force-length ($\mathbf{f}^{PE}$), force-velocity ($\mathbf{f}^V$) and tendon force-length ($\mathbf{f}^T$) dimensionless curves:

$$F^M = F_{max}^M(a\mathbf{f}^l(\tilde{l}^M)\mathbf{f}^V(\tilde{v}^M) + \mathbf{f}^{PE}(\tilde{l}^M)) \tag{2}$$

$$F^T = F_{max}^M\mathbf{f}^T(\tilde{l}^T) \tag{3}$$

where, $F^M$ and $F^T$ are the muscle and tendon forces, respectively. The $F_{max}^M$ parameters for each muscle were computed using regression equations that relate muscle volume to each subject's mass and height [39].

Two types of models were created for each subject. Firstly, functionally-calibrated models (FCMs) were developed using the calibration problem described in the next section. Secondly, non-linearly scaled models (NSMs) were generated using the anthropometric method developed by Modenese et al. [17]. This approach consists in mapping the normalized muscle operating conditions of an existing reference model (i.e., the generic model described above), onto a scaled model of different anthropometric dimensions for equivalent joint configurations.

## Muscle calibration problem formulation

The $l_o^M$ and $l_s^T$ parameters of 18 muscles per leg (specified in S1 Fig in S1 File) were estimated on GPOPS-II [40], using the direct collocation method in a multiphase optimal control problem (OCP). ADiGator [41], a MATLAB automatic differentiation package, was used to improve the computational efficiency of the problem. Three trials per activity were used in the calibration problem, where each phase of the OCP corresponded to a different trial. Since the experimental data of the gait motions were used for testing and evaluation purposes, they were not included in the MT parameters optimization.

Prior to running the muscle calibration problems, the size of the solution space of the $l_o^M$ and $l_s^T$ parameters was constrained following the approach developed in [36]. To do so, physiological combinations of $l_o^M$ and $l_s^T$ parameters were determined by running simple simulations of the ankle, knee and hip joints along their range of motions (ROMs). Since the correlation between the two parameters was approximately linear when representing the feasible combinations as $1/l_o^M$ as a function of $l_s^T/l_o^M$, a linear fit was performed to define the solution space of the parameters [36]:

$$\frac{1}{l_o^M} - c_1\frac{l_s^T}{l_o^M} - c_2 = \delta \tag{4}$$

where $c_1$ and $c_2$ are the coefficients of the line equation and $\delta$ is the deviation from the linear fit. This linear fit allowed for a parameter transformation in the optimal control problem to improve the numerical condition of the problem.

The states of the OCP included the activations and the normalized tendon forces of the muscles, whereas the controls comprised the excitations and the scaled time derivatives of the normalized tendon forces [42]. The static parameters of the OCP included $l_s^T/l_o^M$, $\delta$ and a factor $k_{EMG}$ for each muscle that was used to scale the EMG data as a further normalization of the signals. The activation dynamics was implemented explicitly using (1), whereas the contraction dynamics was modeled implicitly in the problem constraints, following the formulation developed by [42], to improve the efficiency and robustness of the problem. In fact, this simplified the calculation of the nonlinear equations describing muscles' contractions. Additional path constraints bounded: (i) the difference between the muscle-generated joint torques and the inverse dynamics torques ($T_{error}$), (ii) the difference between the modeled muscle excitations and the EMG signals scaled by $k_{EMG}$ ($e_{error}$), (iii) the normalized muscle length to ensure physiological operating conditions of the muscles, (iv) the ratio between the parameters of the right and left muscles to enforce a good level of symmetry, and (v) the discrepancy between the $l_o^M$ of the three vastii and the $l_o^M$ of both gastrocnemii to ensure anatomical accuracy [36]. It is important to note that the torques considered in $T_{error}$ were for the hip flexion/extension (hipFE), hip adduction/abduction (hipAA), hip internal/external rotation (hipIE), knee flexion/extension (kneeFE), ankle plantar flexion/dorsiflexion (anklePD), and subtalar inversion/eversion (subtalarIE). The cost function of the OCP was defined as follows:

$$J = \int_{t_0}^{t_f} \left(w_1(e_{error})^2 + w_2(T_{error})^2 + w_3(a_{nc})^2 - w_4(\Delta_{lower})^2 - w_5(\Delta_{upper})^2\right)dt \tag{5}$$

where $a_{nc}$ is the activation of the non-calibrated muscles, $w_{1-5}$ are weight factors attributed to each cost function term, and $\Delta_{lower}$ and $\Delta_{upper}$ are the differences between $1/l_o^M$ and the bounds of the solution space previously defined. The terms that were added in the cost function were minimized by the OCP, whereas the subtracted ones were maximized. $e_{error}$ and $T_{error}$ were also included in the cost function to ensure that torque and excitation errors were minimized as much as possible, as well as being bound in the path constraints. $\Delta_{lower}$ and $\Delta_{upper}$ were included to penalize the optimal fiber lengths from being too close to the bounds of the solution space to avoid the generation of high passive forces. Furthermore, the initial guesses for $\delta$ and $l_s^T/l_o^M$ were derived from the MT parameters of the non-linearly scaled model (described in the next section). The initial guess for the $k_{EMG}$ factors was set to 0.8 for all muscles. Fig 1a summarizes the muscle calibration problem formulation in a flowchart.

This muscle calibration problem was considered the baseline calibration and it was used to estimate the subject-specific MT parameters of the baseline calibrated models ("FCM-baseline") of the three subjects. Three variations to the OCP of the baseline calibration were carried out. Firstly, the $\Delta_{lower}$ and $\Delta_{upper}$ terms were removed from the cost function to increase the freedom of the static parameters in the solution space. This calibration problem computed the MT parameters of the calibrated models labeled as "FCM-bounds". Secondly, three gait trials were included in the calibration problem instead of the squat trials. This variation of the baseline OCP determined the MT parameters of the calibrated models hereinafter referred to as "FCM-gait". Thirdly, the last variation involved the removal of the tracking terms of the joint torques in the transverse and frontal planes from the path constraints and cost function. Therefore, only the joint torques in the sagittal plane were tracked in this problem variation. The MT parameters resulting from this OCP were used in the calibrated models labeled as "FCM-sagittal".

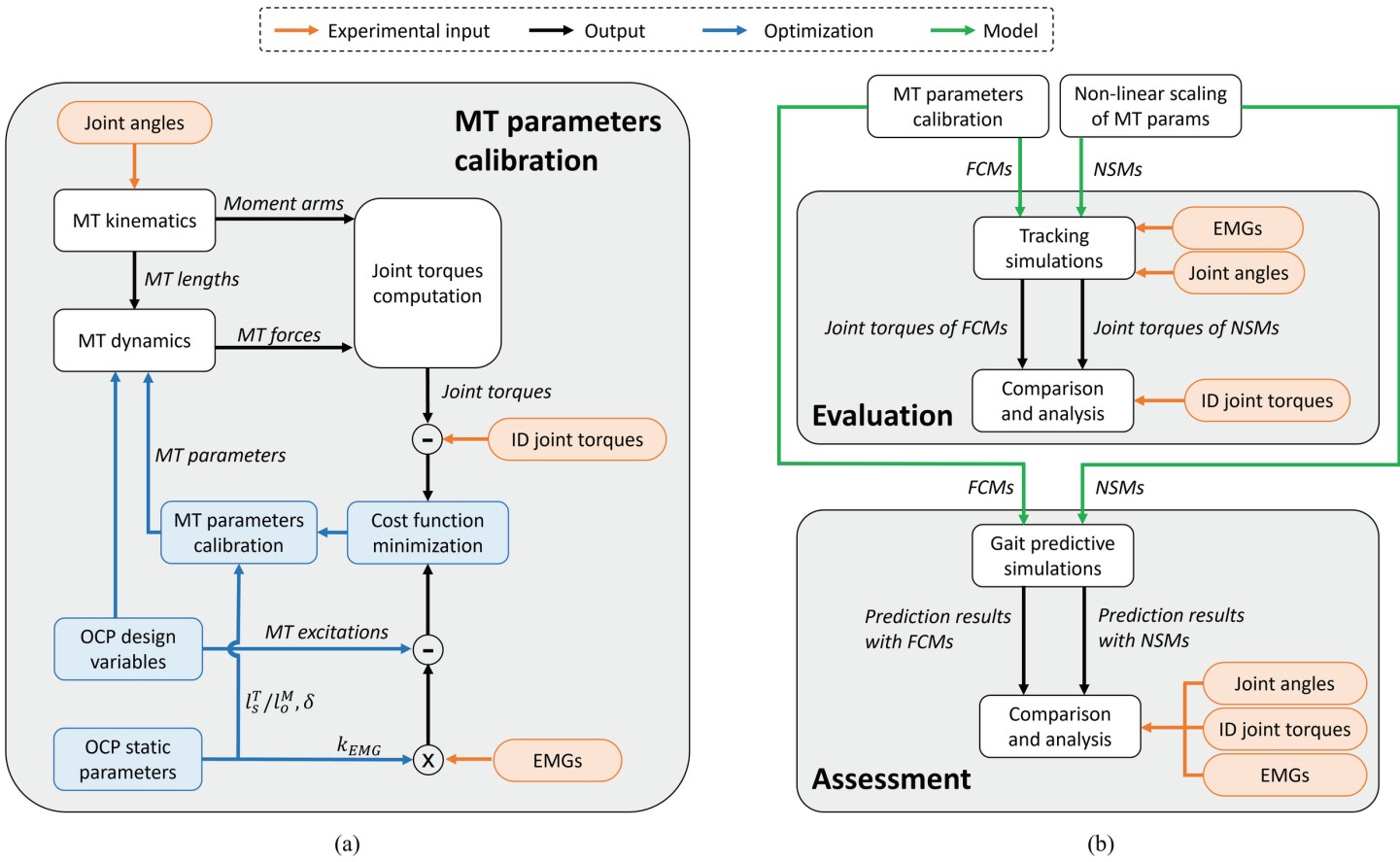

**Fig 1. Flowcharts outlining the MT parameters calibration method, evaluation and assessment of the musculoskeletal models.** (a) A flowchart summarizing the MT parameters calibration process. The subject-specific MT parameters are estimated by minimizing the difference between the model's torques and the ID joint torques, as well as the difference between the excitations of the muscle models and the respective EMG signals. (b) A flowchart outlining the evaluation and assessment of the FCMs in tracking and gait predictive simulations, respectively. The tracking simulations are used to evaluate the capability of the FCMs to estimate joint torques, compared to NSMs. Subsequently, the FCMs and NSMs are assessed in gait predictive simulations. The results from both models are compared against experimental measurements to determine their accuracy.

### Evaluation in tracking simulations

Tracking simulations were performed to evaluate the calibrated models. To perform these simulations, an OCP was formulated similarly to the muscle calibration problem, excluding the static parameters and joint torques tracking terms. For each subject, tracking simulations of 5 trials were carried out, using each version of FCMs (i.e., FCM-baseline, FCM-bounds, FCM-gait and FCM-sagittal). The motion used for the tracking simulations with FCM-baseline, FCM-bounds and FCM-sagittal was gait; whereas, for the simulations with FCM-gait, the squatting activity was tracked. In fact, as previously mentioned, only the motions that were excluded from the calibration were then used for evaluation purposes. Each simulation was repeated with the NSM of the three subjects. It is important to note that, since the estimated $k_{EMG}$ parameters changed in each calibration problem variation, the EMG signals input in the tracking simulations were scaled differently when using a different version of FCMs. To distinguish each tracking simulation in the results section, hereinafter, they will be referred to as TS-baseline, TS-bounds, TS-gait and TS-sagittal based on the FCM variation that was used. Even though the NSMs remained unchanged in the different tracking

simulations, the variation in the scaling of the EMG signals led to differences in the estimated torques.

Subsequently, the coefficient of determination ($R^2$) and the normalized root mean square error (nRMSE) between the ID torques and the models' torques were computed for the NSMs and FCMs in each tracking simulation. The root mean square error was normalized by dividing it by the maximum value of the corresponding ID torque. This analysis was performed for the following DOFs in both legs: hipFE, hipAA, hipIE, kneeFE, anklePD and subtalarIE. Then, we performed a statistical analysis to examine whether there was a significant difference in the nRMSE and $R^2$ values at each DOF between FCMs and NSMs. To do so, we carried out a Mann-Whitney U-test, considering a p-value of less than 0.05 as indicative of a statistically significant difference.

### Assessment in gait predictive simulations

Gait predictive simulations were conducted to test the FCMs against the NSMs. Once again, GPOPS-II was used to run the OCP with direct collocation to perform these simulations. Since the EMG signals were not input into the problem, the activation dynamics was expressed implicitly, as a set of linear equality and inequality constraints [43]. This allowed us to remove muscle excitations from the design variables of the OCP and improve its computational efficiency. For the post-simulation analysis, the muscle excitations were computed from the muscle activations and their derivatives using (1). The formulation of the OCP was similar to the one described in [7]. A detailed explanation of the formulation is included in the supplementary material (S1 File).

All FCM variations and NSMs were tested in the gait predictive simulations. The accuracy of the predicted results was then assessed by comparing them with experimental data. Similarly to the tracking simulations, $R^2$ and nRMSEs were computed between the estimated joint torques and the ID torques. Additionally, the predicted muscle excitations were compared to the EMG signals measured during the gait trials. Since Zajac's first-order activation dynamics may not fully capture the complex, non-linear transitions between muscle activation and deactivation, we limited the analysis on comparing the timing and synchronization between the predicted excitations of the MT actuators and the EMG signals. To this end, we computed the cross-correlation between them for all calibrated muscles. The same statistical analysis used for the tracking simulations was applied to all these quantities as well. Furthermore, the predicted kinematics was compared to the experimental data. More specifically, we performed a qualitative comparison between the experimental and the predicted joint angles, by plotting them on the same graphs. Fig 1b summarizes the evaluation and assessment procedures of the calibrated models.

## Results

### Tracking simulations

FCM-baseline led to significantly lower nRMSEs between the ID and estimated torques in the gait tracking simulations in four DOFs (Fig 2a), compared to NSMs. However, the kneeFE torques were more accurately computed when using NSMs. The $R^2$ values between the ID and the models' torques were significantly higher when using FCM-baseline, for anklePD and hipAA (Fig 2b). Another version of Fig 2 with root mean square error values divided by body mass is shown in S2 Fig in S1 File. Similar conclusions can be drawn with FCM-bounds, which also yielded significantly lower nRMSEs in four DOFs compared to NSMs. In this case, lower nRMSEs were achieved for subtalarIE of FCM-bounds, whereas the

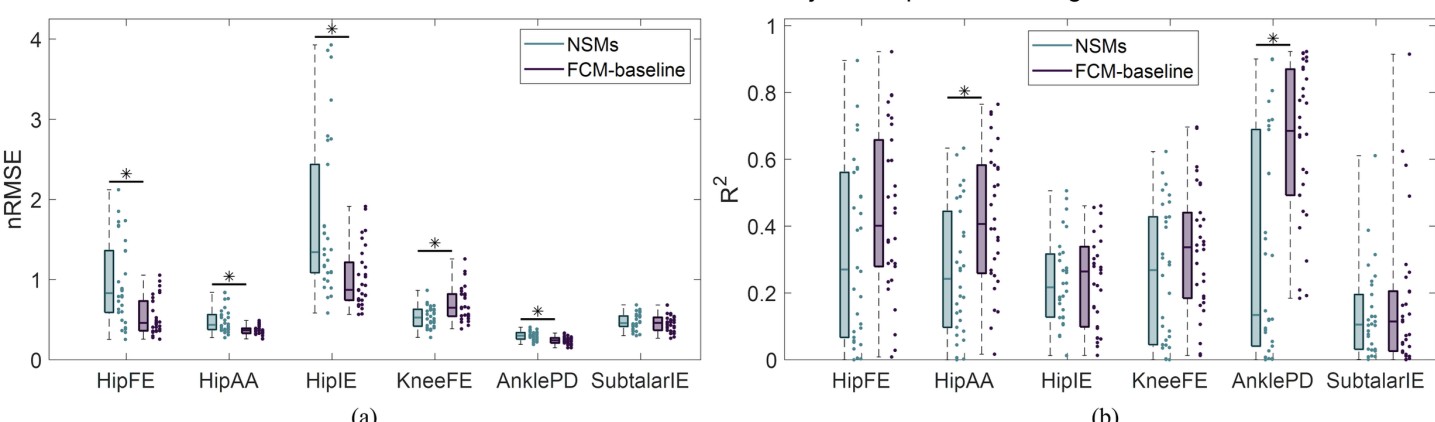

**Fig 2. (a) Box plots of the nRMSEs between the ID and estimated joint torques, when using FCM-baseline and NSMs in gait tracking simulations.** The dots on the right-hand side of each box plot indicate the individual nRMSE values for the right and left legs of each subject. An asterisk is shown if the difference between the nRMSEs of NSMs and FCM-baseline is statistically significant. (b) Box plots of the $R^2$ values between the ID and estimated joint torques, when using FCM-baseline and NSMs in gait tracking simulations. The dots on the right-hand side of each box plot indicate the individual $R^2$ values for the right and left legs of each subject. An asterisk is shown if the difference between the $R^2$ of NSMs and FCM-baseline is statistically significant.

hipAA torques were not significantly different between the two models (Table 2). It is also worth noting that, in the TS-bounds, the standard deviations of the nRMSEs are smaller for both models, indicating that these simulations experienced lower error variability. The $R^2$ values for the hipAA, anklePD and subtalarIE torques of FCM-bounds were significantly higher compared to NSMs. Additionally, in the squat simulations (TS-gait), FCM-gait once again led to lower nRMSEs of torques in four DOFs. Significant difference in $R^2$ values of TS-gait was only achieved for the kneeFE torques, that were more accurately computed using FCM-gait. Moreover, FCM-sagittal performed the worst compared to the other FCMs. In fact, it yielded lower nRMSEs than the NSMs in only three DOFs, and it generated less accurate torques in the frontal and transverse planes, compared to the other FCMs in the other gait tracking simulations (TS-baseline and TS-bounds). The joint torque plots over the gait cycle are included in S1 File (S3–S6 Figs).

**Table 2. Mean ± standard deviation values of nRMSEs and $R^2$ between the ID and estimated torques in the tracking simulations. The values highlighted in bold indicate statistically significant difference between NSMs and the FCMs.**

| | | TS-baseline | | TS-bounds | | TS-gait | | TS-sagittal | |
|---|---|---|---|---|---|---|---|---|---|
| | | NSMs | FCM-baseline | NSMs | FCM-bounds | NSMs | FCM-gait | NSMs | FCM-sagittal |
| HipFE | nRMSE | **0.96 ± 0.52** | **0.55 ± 0.23** | **0.68 ± 0.28** | **0.47 ± 0.16** | **0.59 ± 0.34** | **0.39 ± 0.29** | **1.39 ± 0.97** | **0.93 ± 0.57** |
| | $R^2$ | 0.32 ± 0.27 | 0.44 ± 0.25 | 0.46 ± 0.23 | 0.48 ± 0.25 | 0.75 ± 0.24 | 0.83 ± 0.15 | 0.24 ± 0.21 | 0.28 ± 0.25 |
| HipAA | nRMSE | **0.49 ± 0.15** | **0.37 ± 0.06** | 0.42 ± 0.11 | 0.43 ± 0.08 | **1.36 ± 0.61** | **1.00 ± 0.40** | 1.25 ± 0.88 | 0.90 ± 0.73 |
| | $R^2$ | **0.27 ± 0.19** | **0.42 ± 0.21** | **0.33 ± 0.21** | **0.45 ± 0.21** | 0.34 ± 0.23 | 0.30 ± 0.27 | **0.29 ± 0.19** | **0.42 ± 0.19** |
| HipIE | nRMSE | **1.72 ± 0.98** | **1.02 ± 0.38** | **1.33 ± 0.42** | **0.66 ± 0.33** | **3.17 ± 1.00** | **2.21 ± 0.94** | **3.29 ± 1.81** | **2.28 ± 1.20** |
| | $R^2$ | 0.24 ± 0.13 | 0.24 ± 0.14 | 0.22 ± 0.12 | 0.26 ± 0.15 | 0.51 ± 0.29 | 0.47 ± 0.30 | 0.27 ± 0.16 | 0.33 ± 0.16 |
| KneeFE | nRMSE | **0.53 ± 0.13** | **0.70 ± 0.21** | **0.45 ± 0.12** | **0.71 ± 0.19** | **0.59 ± 0.12** | **0.31 ± 0.11** | **0.53 ± 0.16** | **0.67 ± 0.19** |
| | $R^2$ | 0.25 ± 0.20 | 0.33 ± 0.19 | 0.3 ± 0.22 | 0.37 ± 0.21 | **0.09 ± 0.09** | **0.49 ± 0.29** | 0.26 ± 0.21 | 0.32 ± 0.20 |
| AnklePD | nRMSE | **0.30 ± 0.05** | **0.24 ± 0.05** | **0.29 ± 0.03** | **0.22 ± 0.03** | 0.86 ± 0.71 | 1.18 ± 0.81 | **0.31 ± 0.07** | **0.25 ± 0.06** |
| | $R^2$ | **0.32 ± 0.32** | **0.65 ± 0.24** | **0.47 ± 0.30** | **0.79 ± 0.16** | 0.30 ± 0.22 | 0.26 ± 0.20 | **0.31 ± 0.31** | **0.58 ± 0.29** |
| SubtalarIE | nRMSE | 0.47 ± 0.10 | 0.46 ± 0.11 | **0.48 ± 0.11** | **0.41 ± 0.10** | **0.46 ± 0.22** | **0.74 ± 0.38** | 0.47 ± 0.11 | 0.46 ± 0.10 |
| | $R^2$ | 0.14 ± 0.13 | 0.17 ± 0.22 | **0.13 ± 0.14** | **0.27 ± 0.25** | 0.25 ± 0.24 | 0.32 ± 0.28 | 0.14 ± 0.14 | 0.16 ± 0.20 |

## Gait predictive simulations

The nRMSEs between the ID and predicted torques in the gait predictive simulations were significantly lower for NSMs, compared to FCM-baseline, in four DOFs: hipAA, hipIE, kneeFE and subtalarIE (Fig 3a). Significantly higher $R^2$ values were achieved by NSMs for the hipAA, anklePD and subtalarIE torques (Fig 3b). Another version of Fig 3 with root mean square error values divided by body mass is shown in S7 Fig in S1 File. FCM-bounds performed slightly better compared to FCM-baseline, with improved predictions for its hipIE and kneeFE torques, which were not significantly different from those of NSMs (Table 3). This improvement was also reflected by the $R^2$ values, which suggested that only the subtalarIE torque was predicted more accurately by NSMs. In addition, with respect to FCM-baseline, FCM-gait yielded more accurate joint torques predictions as well. More specifically, the nRMSEs of hipIE and kneeFE decreased when employing FCM-gait. FCM-sagittal led to lower nRMSEs in hipAA, kneeFE and subtalar torques and higher $R^2$ values in anklePD and

### nRMSE and R² values between ID and estimated joint torques in predictive simulations

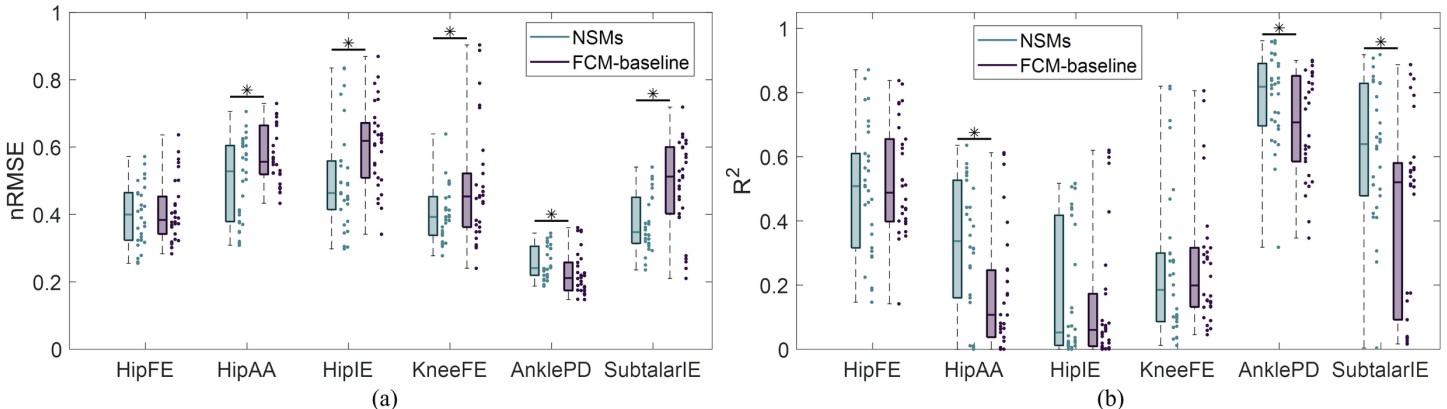

**Fig 3. (a) Box plots of the nRMSEs between the ID and estimated joint torques, when using FCM-baseline and NSMs in gait predictive simulations.** The dots on the right-hand side of each box plot indicate the individual nRMSE values for the right and left legs of each subject. An asterisk is shown if the difference between the nRMSEs of NSMs and FCM-baseline is statistically significant. (b) Box plots of the $R^2$ values between the ID and estimated joint torques, when using FCM-baseline and NSMs in gait predictive simulations. The dots on the right-hand side of each box plot indicate the individual $R^2$ values for the right and left legs of each subject. An asterisk is shown if the difference between the $R^2$ of NSMs and FCM-baseline is statistically significant.

**Table 3. Mean $\pm$ standard deviation values of nRMSEs and $R^2$ between the ID and estimated torques in the gait predictive simulations. The values for the FCMs highlighted in bold indicate a statistically significant difference from NSMs.**

| | | NSMs | FCM-baseline | FCM-bounds | FCM-gait | FCM-sagittal |
|---|---|---|---|---|---|---|
| HipFE | nRMSE | $0.40 \pm 0.09$ | $0.41 \pm 0.09$ | $0.38 \pm 0.05$ | $0.40 \pm 0.09$ | $0.42 \pm 0.12$ |
| | $R^2$ | $0.50 \pm 0.20$ | $0.52 \pm 0.17$ | $0.47 \pm 0.18$ | $0.52 \pm 0.21$ | $0.45 \pm 0.26$ |
| HipAA | nRMSE | $0.50 \pm 0.12$ | $\mathbf{0.57 \pm 0.08}$ | $\mathbf{0.62 \pm 0.06}$ | $\mathbf{0.60 \pm 0.08}$ | $0.49 \pm 0.16$ |
| | $R^2$ | $0.32 \pm 0.21$ | $\mathbf{0.18 \pm 0.19}$ | $0.38 \pm 0.23$ | $\mathbf{0.21 \pm 0.23}$ | $0.37 \pm 0.26$ |
| HipIE | nRMSE | $0.50 \pm 0.15$ | $\mathbf{0.61 \pm 0.13}$ | $0.53 \pm 0.12$ | $0.53 \pm 0.19$ | $\mathbf{0.83 \pm 0.33}$ |
| | $R^2$ | $0.17 \pm 0.20$ | $0.14 \pm 0.20$ | $0.20 \pm 0.18$ | $0.16 \pm 0.22$ | $0.20 \pm 0.21$ |
| KneeFE | nRMSE | $0.40 \pm 0.08$ | $\mathbf{0.48 \pm 0.17}$ | $0.39 \pm 0.10$ | $0.43 \pm 0.14$ | $0.41 \pm 0.08$ |
| | $R^2$ | $0.26 \pm 0.24$ | $0.26 \pm 0.20$ | $0.36 \pm 0.26$ | $0.27 \pm 0.25$ | $0.29 \pm 0.21$ |
| AnklePD | nRMSE | $0.26 \pm 0.05$ | $\mathbf{0.22 \pm 0.06}$ | $\mathbf{0.20 \pm 0.04}$ | $0.24 \pm 0.07$ | $\mathbf{0.22 \pm 0.04}$ |
| | $R^2$ | $0.78 \pm 0.14$ | $\mathbf{0.70 \pm 0.16}$ | $0.80 \pm 0.11$ | $\mathbf{0.71 \pm 0.13}$ | $0.71 \pm 0.15$ |
| SubtalarIE | nRMSE | $0.38 \pm 0.09$ | $\mathbf{0.49 \pm 0.14}$ | $\mathbf{0.45 \pm 0.16}$ | $\mathbf{0.46 \pm 0.10}$ | $0.43 \pm 0.17$ |
| | $R^2$ | $0.61 \pm 0.22$ | $\mathbf{0.43 \pm 0.30}$ | $\mathbf{0.28 \pm 0.32}$ | $\mathbf{0.45 \pm 0.33}$ | $\mathbf{0.31 \pm 0.31}$ |

hipAA torques, compared to FCM-baseline. The predicted joint torque profiles against the gait cycle are shown in S1 File (S8–S11 Figs).

No significant differences were found between NSMs and FCM-baseline regarding the cross-correlation between the EMG signals and the MT models excitations (S12 Fig in S1 File). FCM-bounds and FCM-gait, on the other hand, generated significantly less accurate predictions of excitations for some of the muscles (Table 4). As opposed to FCM-baseline, FCM-sagittal led to higher correlation between the glutei and biceps femoris excitations and the respective EMG signals.

Furthermore, the kinematics was predicted more accurately by NSMs, compared to FCM-baseline (Fig 4). In particular, the FCMs consistently under-flexed the knee during swing. In fact, in subjects 1 and 2, the kneeFE angles of FCM-baseline were missing the characteristic peak observed in both the experimental data and the simulations using NSMs. NSMs and FCM-baseline had difficulty in accurately predicting the hipAA angle for subjects 2 and 3, whereas, for subject 1, NSMs predicted it more accurately than FCM-baseline. Additionally, both models consistently overestimated the ankle plantar flexion at the beginning of the gait cycle. In general, the most-accurately predicted angle by both models was the hipFE angle. Nevertheless, in subjects 1 and 2, the left hipFE profiles of the NSMs were closer to the experimental data, compared to FCM-baseline. The predicted kinematics using FCM-bounds, FCM-gait and FCM-sagittal are shown in the supplementary material (S13–S15 Figs in S1 File).

## Discussion

A MT-parameters calibration problem was developed with optimal control techniques, evaluated using motion tracking simulations and assessed in fully-predictive simulations of gait. Three different variations to the baseline calibration problem were investigated for three healthy subjects. More specifically, we explored how (i) deducting the penalisation of the solution space near the bounds of the MT parameters, (ii) adding gait to the calibration motions, and (iii) tracking only the joint torques in the sagittal plane affected the FCMs in tracking and predictive simulations compared to the NSMs. In general, the results suggest that, while the FCMs yielded more accurate joint torques estimations in the tracking simulations, they did not outperform the scaled models in the gait predictions.

When comparing the gait predictions of FCM-baseline and NSMs, it is clear that the resultant kinematics significantly influenced the results of the joint torques. In fact, in the DOFs where the predicted kinematics was considerably different from the experimental data, a higher nRMSE was found for the corresponding torques. For instance, since NSMs predicted

**Table 4. Mean ± standard deviation values of cross-correlations between the EMG signals and the estimated excitations in the gait predictive simulations. The values for the FCMs highlighted in bold indicate a statistically significant difference from NSMs.**

|                       | NSMs        | FCM-baseline | FCM-bounds      | FCM-gait        | FCM-sagittal    |
|-----------------------|-------------|--------------|-----------------|-----------------|-----------------|
| Glutei                | 0.48 ± 0.16 | 0.47 ± 0.16  | 0.48 ± 0.19     | 0.51 ± 0.16     | **0.53 ± 0.13** |
| Vasti                 | 0.50 ± 0.19 | 0.48 ± 0.22  | **0.43 ± 0.17** | 0.45 ± 0.23     | 0.50 ± 0.20     |
| Rectus femoris        | 0.63 ± 0.15 | 0.56 ± 0.18  | **0.47 ± 0.19** | **0.49 ± 0.17** | 0.57 ± 0.14     |
| Semimem. and semiten. | 0.41 ± 0.19 | 0.35 ± 0.20  | **0.29 ± 0.22** | 0.42 ± 0.23     | 0.40 ± 0.18     |
| Biceps femoris        | 0.40 ± 0.15 | 0.44 ± 0.19  | **0.33 ± 0.18** | 0.44 ± 0.14     | **0.47 ± 0.17** |
| Gastrocnemii          | 0.41 ± 0.13 | 0.36 ± 0.20  | **0.34 ± 0.17** | **0.34 ± 0.16** | 0.36 ± 0.19     |
| Soleus                | 0.29 ± 0.11 | 0.33 ± 0.18  | 0.27 ± 0.16     | 0.27 ± 0.14     | 0.32 ± 0.16     |
| Tibialis anterior     | 0.38 ± 0.17 | 0.33 ± 0.20  | **0.20 ± 0.11** | 0.33 ± 0.17     | 0.36 ± 0.19     |

## Predicted and experimental joint angles

**Fig 4. The predicted hipFE, hipAA, hipIE, kneeFE, anklePD and subtalarIE angles of NSMs and FCM-baseline, compared to the respective experimental joint angles for the three subjects.**

more accurately the kneeFE angles compared to FCM-baseline, the latter led to significantly higher nRMSEs in kneeFE torques. Similar conclusions can be drawn from the hipAA DOF because NSMs yielded more accurate kinematics for subject 1. On the other hand, when the kinematics was predicted similarly by both models, there were either no significant differences between them in the corresponding torque nRMSE values (i.e., hipFE) or FCM-baseline yielded significantly lower nRMSE values (i.e., anklePD). Furthermore, the inaccuracy of the predicted knee kinematics of FCM-baseline was caused by high passive forces in the vastus medialis (for subjects 1 and 2) and in the vastus intermedius (for subject 3) during knee flexion. Falisse et al. [7] reported similar findings, as they also found that elevated passive muscle forces negatively impacted the knee angles estimations during gait predictions. In fact, when they included a term in the cost function to minimize the passive joint torques, they could more accurately predict knee extension. In our work, the high forces arose from abnormally low $l_s^T$ values in the vastus medialis and vastus intermedius of FCM-baseline. These findings highlight the importance of choosing appropriate and anatomically-informed bounds for the MT parameters in the muscle calibration problems. Future studies should consider tighter limits for the $l_s^T$ values or, alternatively, adding another path constraint to constrain similarity between the $l_s^T$ values of the three vasti muscles.

Since the first variation of the calibration problem allowed for more freedom in the solution space of the MT parameters, some $l_o^M$ and $l_s^T$ values estimated by the OCP were close

to the bounds of the parameters. For this reason, the muscles actuating hipAA and hipIE produced high passive forces in the motions used in the calibration. Consequently, in order to compensate and produce smaller active forces, their scale factors for the EMG signals ($k_{EMG}$) were very low. Given that $k_{EMG}$ was then input into the gait tracking simulations, the hipAA and hipIE torques of NSMs in TS-bounds were much smaller compared to TS-baseline. This caused the nRMSEs to be much smaller in TS-bounds, because the high $k_{EMG}$ factors in TS-baseline provoked a significant overestimation of joint torques in NSMs (Table 2). A few discussion points can be drawn from this. Firstly, as mentioned in the previous paragraph, it is fundamental to bound the MT parameters in the calibration problems to prevent an over- or under-generation of passive forces. To this end, the values of passive forces should be thoroughly checked for each motion in the calibration problem to ensure that they are physiologically reasonable. Alternatively, other constraints could be added for the normalized muscle lengths to ensure that the muscles work within their realistic operating condition. For instance, in the muscle calibration problem developed by Falisse et al. [32], the normalized muscle lengths were enforced to cross their optimal fiber lengths during motions. Secondly, the normalization of the EMG signals considerably influenced the estimation of the MT parameters. However, it is extremely difficult to accurately normalize the signals because maximum muscle contractions are rarely reached in practice. Even though we added the $k_{EMG}$ factor as an optimization parameter to mitigate this issue, it was challenging to ensure that these EMG profiles "scaled" by $k_{EMG}$ were realistic. Thirdly, as stated in [44], the maximum isometric force parameters in the generic OpenSim models are usually an overestimation of the real values. Although we computed the maximum isometric forces using the regression model described in [39], some muscles exhibited values that exceeded those in the generic model developed by Catelli et al. [38]. Therefore, excessively high maximum isometric forces could exaggerate the aforementioned effects of abnormal muscle passive forces and inaccurate normalization of EMG signals. All of these points will be addressed in future work to ensure that the calibration process estimates accurate and realistic MT parameters.

Differently from FCM-baseline, the muscles excitations generated by FCM-bounds in the gait predictions had significantly lower cross-correlation values compared to NSMs (Table 4). In fact, since the MT parameters of FCM-bounds were closer to their upper or lower bounds, the activation timings of some muscles were also not realistic. For instance, since the $l_o^M$ values of the vasti muscles in FCM-bounds were larger than the ones in NSMs and FCM-baseline, these actuators had to activate for a longer time during the beginning of swing. Similarly, the $l_o^M$ values of the tibialis anterior were also larger for FCM-bounds, compared to NSMs and FCM-baseline. This triggered FCM-bounds to activate the tibialis anterior too early in the swing phase during the plantar flexion of the ankle. Therefore, these results suggest that, even though the EMG signals are tracked during the muscle calibration problem, giving too much freedom to the MT parameters might cause erroneous activation timings in gait predictive simulations.

Falisse et al. [26] found that the inclusion of gait as a motion in the calibration problem is fundamental to estimate accurate MT parameters. In that particular study, the authors focused their analysis on the kneeFE torque and the muscles actuating that joint. Our results agree with their statement, as FCM-gait produced significantly lower nRMSE and higher $R^2$ values for kneeFE torques compared to NSMs in the tracking simulations—a trend that was not observed with FCM-baseline. Even though the tracking simulations with FCM-gait were carried out with squat data as opposed to gait, the significant difference in nRMSEs and $R^2$ values between FCM-gait and NSMs supports the statement raised by Falisse et al. [26] (Table 2). In addition, in the gait predictive simulations, the use of FCM-gait improved the predictions of the kneeFE torques with respect to FCM-baseline. In fact, there were

no significant differences in kneeFE torques between FCM-gait and NSMs; whereas, the corresponding nRMSE of FCM-baseline was significantly lower than that of NSMs. One possible explanation is that the predicted kinematics using FCM-gait were slightly more similar to the experimental data compared to FCM-baseline for subjects 1 and 3 (S14 Fig in S1 File).

As expected, compared to the baseline calibration, the calibration problem that did not track the joint torques in the frontal and transverse plane led to significantly different MT parameters of some muscles that mainly operate in these two planes (S2 Table). For instance, the subject-specific MT parameters of the gluteus medius actuators (the prime movers of hip abduction) changed substantially between FCM-baseline and FCM-sagittal. Consequently, the $k_{EMG}$ factors for these muscles estimated in the third variation of the calibration problem were higher than those in the baseline calibration. This caused high nRMSEs in hipAA and hipIE torques for both NSMs and FCM-sagittal in TS-sagittal. In the predictive simulations, FCM-sagittal unexpectedly led to lower nRMSE and higher $R^2$ values for hipAA torques, compared to FCM-baseline (Table 3). In fact, when using FCM-sagittal, as opposed to FCM-baseline, the predicted hipAA angles were more similar to the experimental data for subjects 1 and 3 (S15 Fig in S1 File). As mentioned above, the resultant kinematics considerably affects the prediction of the joint torques. In addition, FCM-sagittal is the only model that yielded a significantly higher correlation between the excitations of the glutei and the corresponding EMG signals, compared to NSMs. These results suggest that the substantial difference between the glutei parameters in FCM-sagittal and the other FCMs had a surprisingly positive effect on the estimation of the glutei excitations, as well as the hipAA angles and torques in the gait predictive simulations. These discrepancies between the prediction results of FCM-sagittal and the other FCMs might be caused by the presence of multiple local minima in the OCP of the predictive simulations. Therefore, changes in the model could lead to considerable variations in the predictions outcomes, which may not necessarily align with the results of the tracking simulations. Regarding the computational time, while the baseline calibration took on average 41 minutes to run, removing the tracking of the torques in the frontal and transverse planes reduced the time to 37 minutes. Therefore, we considered the difference in computational time to be negligible between the two muscle calibration problems.

A few general insights can be drawn from this study. Firstly, it is not clear whether tracking simulations are enough to evaluate the FCMs. If the long-term goal is to employ these models in human motion predictions to explore "what-if" scenarios, they should also be tested in such simulations. In fact, we found that even though in most tracking simulations the FCMs generally yielded significantly lower nRMSE and higher $R^2$ values for joint torques compared to the scaled models, we did not observe such a significant difference in the gait predictive simulations. Therefore, the solutions of the predictions were less sensitive to the MT parameters than in the tracking simulations. The optimality criteria in the cost functions and the constraints set in the gait predictions had a more significant influence on the OCP solutions, compared to the MT parameters estimation approach. Secondly, this work highlights the importance of using multiple subjects in computational biomechanics studies. For instance, if we only analyzed subject 3, we would have not found a substantial discrepancy between the predicted kinematics of the FCMs and the NSMs (Fig 4). Thirdly, the considerable variations in MT parameters generated by the different calibration problems highlight the high number of possible solutions for the parameters. Therefore, it is possible that the OCPs used for the MT parameters calibrations contain many local minima. For this reason, it was difficult to ensure that a global minimum was found in the calibration problems and that the

estimated static parameters represented the "real" subject-specific MT parameters. This reinforces the necessity of setting stricter physiological bounds in the OCP to reduce the solution space of the static parameters and the likelihood of finding local minima. Lastly, even though this study suggests that non-linear scaling of musculoskeletal models may be sufficient for gait predictive simulations of healthy subjects, a proper MT parameters calibration may be fundamental when carrying out these simulations for subjects with neuromuscular diseases or traumatic injuries.

This work has some limitations. First, in the muscle calibration problems and the tracking simulations, we did not track the EMG signals of all the muscles included in the OpenSim models due to lack of experimental data. We tried, in fact, to calibrate as many muscles as we could in both lower limbs, using a conventional experimental setup with 16 EMG channels. Therefore, we took a different strategy compared to other studies (e.g., [25,26]), where they measured all the EMG signals needed to personalize and evaluate the parameters of all MT actuators in a joint or in a single leg (with some assumptions). By not doing so, in this study, we could not entirely make sure that the non-calibrated actuators were activated realistically in the muscle calibration problems and the tracking simulations. For this reason, the contribution of the non-measured muscles in the model likely affected both the results of the MT parameters calibration and the joint torques estimation in the evaluation procedures. Second, we used Zajac's activation dynamics [9] which does not take into account the electromechanical delay between the neural excitations and muscle activations. Future research should consider analyzing the effects of using a more complex model (e.g., [45]) and calibrating the parameters of the activation dynamics (e.g., activation/de-activation time constants, electromechanical delay, etc.). Third, the OCPs in this study were run only once using a single initial guess. Multiple sets of initial guesses should be employed to ensure that the problem converged in a global minimum and found the most optimal solution. Even though Falisse et al. [26] reported that the cost function values of their muscle calibration problems remained similar when using an arbitrary guess or a pre-computed hot start, this analysis should have been conducted in our study as well to test the robustness of our problem formulation. Lastly, a higher number of subjects could have led to a more conclusive and convincing set of results. Despite these limitations, the conclusions drawn from this study highlight interesting differences between the applications of NSMs and FCMs in tracking and fully-predictive simulations of human gait.

## Conclusion

In this study, we developed an OCP to calibrate the MT parameters of three healthy subjects, evaluated the resulting models through motion tracking simulations, and assessed them in gait predictive simulations against anthropometrically-scaled models. In addition, since previous studies using functional calibration methods had some distinctions in their approaches, we explored three variations of the baseline calibration problem to investigate these differences. Our results suggest that in the tracking simulations, the FCMs generally led to significantly lower nRMSE and higher $R^2$ values between the experimental and estimated torques, compared to the NSMs. However, in predictive simulations, the FCMs did not provide greater accuracy compared to the NSMs. As future work, we will expand our MT calibration pipeline to personalise parameters related to the activation dynamics and the MT stiffness. Moreover, we will apply these techniques to generate subject-specific musculoskeletal models of spinal cord injury patients to predict the salient traits of the impaired gait patterns of specific individuals.

## Supporting information

**S1 File. Supplementary material.** A PDF file including some details regarding the optimal control problem for the gait predictive simulations and additional figures.
(PDF)

**S2 File. Experimental data and simulation results.**
(CSV)

**S1 Table. Extended version of Table 1.**
(XLSX)

**S2 Table. The MT parameters of all the musculoskeletal models.**
(XLSX)

## Author contributions

**Conceptualization:** Filippo Maceratesi, Míriam Febrer-Nafría, Josep M. Font-Llagunes.

**Data curation:** Filippo Maceratesi.

**Formal analysis:** Filippo Maceratesi.

**Funding acquisition:** Josep M. Font-Llagunes.

**Investigation:** Filippo Maceratesi.

**Methodology:** Filippo Maceratesi, Míriam Febrer-Nafría.

**Project administration:** Josep M. Font-Llagunes.

**Resources:** Filippo Maceratesi, Míriam Febrer-Nafría, Josep M. Font-Llagunes.

**Software:** Filippo Maceratesi.

**Supervision:** Míriam Febrer-Nafría, Josep M. Font-Llagunes.

**Validation:** Filippo Maceratesi.

**Visualization:** Filippo Maceratesi.

**Writing – original draft:** Filippo Maceratesi.

**Writing – review & editing:** Míriam Febrer-Nafría, Josep M. Font-Llagunes.

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
