## [Decision Letter · Decision Letter 0]

3 Apr 2025

PONE-D-24-57517Calibrated muscle models improve tracking simulations without enhancing gait predictionsPLOS ONE

Dear Dr. Maceratesi,

Thank you for submitting your manuscript to PLOS ONE. After careful consideration, we feel that it has merit but does not fully meet PLOS ONE’s publication criteria as it currently stands. Therefore, we invite you to submit a revised version of the manuscript that addresses the points raised during the review process.

We look forward to receiving your revised manuscript.

Kind regards,

Xianlian Zhou

Academic Editor

PLOS ONE

**Journal Requirements:**

1. When submitting your revision, we need you to address these additional requirements. Please ensure that your manuscript meets PLOS ONE's style requirements, including those for file naming. The PLOS ONE style templates can be found at https://journals.plos.org/plosone/s/file?id=wjVg/PLOSOne_formatting_sample_main_body.pdf and https://journals.plos.org/plosone/s/file?id=ba62/PLOSOne_formatting_sample_title_authors_affiliations.pdf 2. Please note that PLOS ONE has specific guidelines on code sharing for submissions in which author-generated code underpins the findings in the manuscript. In these cases, we expect all author-generated code to be made available without restrictions upon publication of the work. Please review our guidelines at https://journals.plos.org/plosone/s/materials-and-software-sharing#loc-sharing-code and ensure that your code is shared in a way that follows best practice and facilitates reproducibility and reuse. 3. Thank you for stating in your Funding Statement: This work was supported in part by Grant PID2021-123657OB-C33 funded by MCIN/AEI/10.13039/501100011033 and by ERDF ”A way of making Europe”. It was also supported in part by the predoctoral program AGAUR-FI ajuts (2022 FI-B 00911) Joan Oró, which is backed by the Secretariat of Universities and Research of the Department of Research and Universities of the Generalitat of Catalonia, as well as the European Social Plus Fund. Please provide an amended statement that declares *all* the funding or sources of support (whether external or internal to your organization) received during this study, as detailed online in our guide for authors at http://journals.plos.org/plosone/s/submit-now.  Please also include the statement “There was no additional external funding received for this study.” in your updated Funding Statement. Please include your amended Funding Statement within your cover letter. We will change the online submission form on your behalf. 4. Thank you for stating the following in the Acknowledgments Section of your manuscript: This work was supported in part by Grant PID2021-123657OB-C33 funded by 539MCIN/AEI/10.13039/501100011033 and by ERDF ”A way of making Europe”. It was 540also supported in part by the predoctoral program AGAUR-FI ajuts (2022 FI-B 00911) 541Joan Or´o, which is backed by the Secretariat of Universities and Research of the 542Department of Research and Universities of the Generalitat of Catalonia, as well as the 543European Social Plus Fund. We note that you have provided funding information that is not currently declared in your Funding Statement. However, funding information should not appear in the Acknowledgments section or other areas of your manuscript. We will only publish funding information present in the Funding Statement section of the online submission form. Please remove any funding-related text from the manuscript and let us know how you would like to update your Funding Statement. Currently, your Funding Statement reads as follows: This work was supported in part by Grant PID2021-123657OB-C33 funded by MCIN/AEI/10.13039/501100011033 and by ERDF ”A way of making Europe”. It was also supported in part by the predoctoral program AGAUR-FI ajuts (2022 FI-B 00911) Joan Oró, which is backed by the Secretariat of Universities and Research of the Department of Research and Universities of the Generalitat of Catalonia, as well as the European Social Plus Fund.  Please include your amended statements within your cover letter; we will change the online submission form on your behalf. 5. When completing the data availability statement of the submission form, you indicated that you will make your data available on acceptance. We strongly recommend all authors decide on a data sharing plan before acceptance, as the process can be lengthy and hold up publication timelines. Please note that, though access restrictions are acceptable now, your entire data will need to be made freely accessible if your manuscript is accepted for publication. This policy applies to all data except where public deposition would breach compliance with the protocol approved by your research ethics board. If you are unable to adhere to our open data policy, please kindly revise your statement to explain your reasoning and we will seek the editor's input on an exemption. Please be assured that, once you have provided your new statement, the assessment of your exemption will not hold up the peer review process.

Reviewers' comments:

Reviewer's Responses to Questions

**Comments to the Author**

1. Is the manuscript technically sound, and do the data support the conclusions?

Reviewer #1: Yes

Reviewer #2: Partly

2. Has the statistical analysis been performed appropriately and rigorously? 

Reviewer #1: Yes

Reviewer #2: Yes

3. Have the authors made all data underlying the findings in their manuscript fully available?

Reviewer #1: Yes

Reviewer #2: No

4. Is the manuscript presented in an intelligible fashion and written in standard English?

Reviewer #1: Yes

Reviewer #2: Yes

5. Review Comments to the Author

**Reviewer #1: **It is an interesting and nice study. Calibration of musculoskeletal model is an important work. Authors tested three functional calibration methods and validated through tracking simulation and predictive gait simulation. Below are some suggestions.

1. In lines 187-188, for the nonlinear scaled model, it is suggested to provide some details instead of only referring to a paper.

2. The rational of Eq. (4) needs further explanation.

3. In Eq. (5), the first term, the error is not clearly defined.

4. In lines 211 and 212, what does it mean "modeled implicitly in the path constraints"? please explain "path" constraints and "implicitly" modelled.

5. In Eqs. (2,3), the variable Fm_max is not defined. In addition, is this variable a constant in this study? Why not include Fm_max as a model parameter? Between different subjects, Fm_max is changing.

**Reviewer #2:** SUMMARY

This paper describes an investigation into the effects of calibrating muscle parameters (muscle and tendon lengths) on joint torques and kinematics in a musculoskeletal simulation. Simulations were generated for 3 subjects. Overall, the authors concluded that there was not a notable difference between a typical non-linearly scaled model and the functionally-calibrated models.

MAJOR COMMENTS

Overall, I found this paper to be interesting and well-written. The authors provide a good overview of other similar studies, and provide justification for their chosen study design. The problem of calibrating muscle parameters remains a challenge in musculoskeletal modeling and simulation, and this study will be a useful addition to the literature. However, several clarifications and improvements are needed.

My primary concern regarding this manuscript is that the authors’ conclusions do not seem to fully align with the data presented. Initially, I was also confused about the graphs presented. This is partially because the image quality in the proofs makes the images difficult to read, but it took me a long time to realize that Fig 2 (mean joint torques) does not contain the same information as Fig 4 (joint angle time histories). Clarity could be enhanced by adding titles to the images, since the unitless nRMSE values do not convey any information about the quantity being presented. Also, considering that the joint torques are a key outcome of this study, it would be helpful to include the time history plots for the joint torques (i.e., in the same format as Fig 4), at least as supplementary material. Aside from these misunderstandings, some of the conclusions in the paper do not seem to be well-supported. First, the paper only reports normalized (nRMSE) differences. This may obscure some substantial differences between the models. For example, in Fig 2a, the differences appear very large, but hip internal/external rotation torques are small in magnitude. Conversely, the differences in knee flexion/extension appear small, but may in fact be large in magnitude since knee flexion/extension torques are large during gait. It would enhance clarity to include non-normalized versions of Figs 2 and 3, perhaps as additional supplementary material. Similarly, in the third paragraph of Results > Gait Predictive Simulations, the authors state that “the kinematics was predicted slightly more accurately by NSMs compared to FCM0 (Fig 4).” Looking at Fig 4, I would characterize the differences in knee flexion as dramatically different. Knee flexion is the largest magnitude joint angle in gait, and has the clearest pattern during gait. The results from FCM0 are missing the characteristic peak, differing from both the NSMs and experimental data by what looks to be 30-50 degrees. There are similarly large differences in other joints, including HipAA, Hip IE, and to some extent HipFE. This mischaracterization of the joint angles casts doubt on the authors’ interpretation of the joint torque results in Figs 2 and 3, leading the reader to wonder if similarly dramatic differences are somewhat hidden by the nRMSE calculations.

Another concern is related to the language used to describe the results. The authors provide a good overview of the literature in which similar studies used joint torques to “validate” the musculotendon parameter calibration. Nonetheless, I would argue that the only true gold standard for “validation” of the musculotendon parameters would be direct measurement from medical imaging to calculate the lengths of the muscle fibers and tendons. While the optimization procedure may produce more accurate joint torques, the muscle and tendon lengths may not be closer to the actual lengths in the human subject. I recommend the authors state that the results were “evaluated” by comparing to joint torques, rather than “validated” to avoid confusion.

Related, there are a number of statements in the Discussion and Conclusion that results were “more accurate” or “less accurate” compared to some other quantity. The manuscript would be more clear if the authors stated precisely what was different, e.g., “correlation was significantly higher” or “nRMSE was significantly lower”. Please review the manuscript and address all occurrences of this issue.

The first paragraph of the Introduction should include more references to the literature. There are a number of techniques which can be referred to as “predictive simulations”: direct collocation, reinforcement learning, direct shooting, etc. The authors should include references to other studies as appropriate. In addition, OpenSim now includes the Moco tool to perform direct collocation simulations, but it does not appear that the authors used Moco in this study. Please clarify this in the manuscript.

MINOR COMMENTS

Table 1: It would be helpful to add the last name of the first author for each reference, rather than just the reference number. There is also a typo in the far right column: “paramerters” should be “parameters”

In the first paragraph under “Musculoskeletal Modeling and OpenSim Analyses”, should say “…using the ‘Scale Tool’ in OpenSim…”

First paragraph under Results > Tracking Simulations, first word should be “FCM0”?

I found the abbreviations FCM0, FCM1, etc difficult to remember which abbreviation corresponded to which type of calibration. As a suggestion, consider changing to something like FCM-baseline, FCM-bounds, FCM-gait, and FCM-sagittal. This would increase readability of the manuscript.

In the Discussion, the authors mention that imprecise estimation of moment arms significantly affects joint torque calculations. This is true, but all modeling/simulation studies share this limitation. Also, if you are going through the trouble to use MRI to estimate moment arms, you could also use MRI to estimate muscle fiber length and tendon slack length. Mentioning muscle moment arms as a limitation does not seem to add much to the discussion – recommend to either remove this, or mention that with MRI estimates of several musculotendon parameters could potentially be made more accurate.

The authors cite [8] Delp et al. 2007 for OpenSim, but the recommendations are to cite the more recent Seth A, Hicks JL, Uchida TK, Habib A, Dembia CL, Dunne JJ, et al. (2018) OpenSim: Simulating musculoskeletal dynamics and neuromuscular control to study human and animal movement. PLoS Comput Biol 14(7): e1006223.

I did not see a reference to [28] Delp and Loan 1995 in the main text.

6. PLOS authors have the option to publish the peer review history of their article (what does this mean?). If published, this will include your full peer review and any attached files.

Reviewer #1: **Yes: **Yujiang Xiang

Reviewer #2: No

---

## [Author Response · Author response to Decision Letter 1]

2 May 2025

*A more comprehensible version of this response is included in the submitted "Response to Reviewers.pdf" file.*

Journal Requirements

Thank you for stating in your Funding Statement:

This work was supported in part by Grant PID2021-123657OB-C33 funded by MCIN/AEI/10.13039/501100011033 and by ERDF ”A way of making Europe”. It was also supported in part by the predoctoral program AGAUR-FI ajuts (2022 FI-B 00911) Joan Oró, which is backed by the Secretariat of Universities and Research of the Department of Research and Universities of the Generalitat of Catalonia, as well as the European Social Plus Fund.

Thank you for the clarification. We added the aforementioned sentence to the funding statement. Please find below the updated funding statement:

This work was supported in part by Grant PID2021-123657OB-C33 funded by MCIN/AEI/10.13039/501100011033 and by ERDF ”A way of making Europe”. It was also supported in part by the predoctoral program AGAUR-FI ajuts (2022 FI-B 00911) Joan Oró, which is backed by the Secretariat of Universities and Research of the Department of Research and Universities of the Generalitat of Catalonia, as well as the European Social Fund Plus. There was no additional external funding received for this study.

Thank you for catching that. We removed the funding statement from the “Acknowledgements” section.

When completing the data availability statement of the submission form, you indicated that you will make your data available on acceptance. We strongly recommend all authors decide on a data sharing plan before acceptance, as the process can be lengthy and hold up publication timelines. Please note that, though access restrictions are acceptable now, your entire data will need to be made freely accessible if your manuscript is accepted for publication. This policy applies to all data except where public deposition would breach compliance with the protocol approved by your research ethics board. If you are unable to adhere to our open data policy, please kindly revise your statement to explain your reasoning and we will seek the editor's input on an exemption. Please be assured that, once you have provided your new statement, the assessment of your exemption will not hold up the peer review process.

Thank you for pointing this out. We made all data available in the supplementary material, in the “S2 File.csv” file.

Data Policy Question

Have the authors made all data underlying the findings in their manuscript fully available?

Reviewer #2: No

Thank you for pointing this out. We made all data available in the supplementary material, in the “S2 File.csv” file.

Reviewer #1

In lines 187-188, for the nonlinear scaled model, it is suggested to provide some details instead of only referring to a paper.

Thank you for the suggestion. Please find below the revised text (lines 197-200):

Secondly, non-linearly scaled models (NSMs) were generated using the anthropometric method developed by Modenese et al. [17]. This approach consists in mapping the normalized muscle operating conditions of an existing reference model (i.e., the generic model described above), onto a scaled model of different anthropometric dimensions for equivalent joint configurations.

The rationale of Eq. (4) needs further explanation.

Good point, thank you very much for the comment. Please find below the added explanation for Eq. 4 (lines 214-219):

Prior to running the muscle calibration problems, the size of the solution space of the l_o^M and l_s^T parameters was constrained following the approach developed in [36]. To do so, physiological combinations of l_o^M and l_s^T parameters were determined by running simple simulations of the ankle, knee and hip joints along their range of motions (ROMs). Since the correlation between the two parameters was approximately linear when representing the feasible combinations as 1/l_o^M as a function of l_s^T/l_o^M, a linear fit was performed to define the solution space of the parameters [36]:

1/(l_o^M )-c_1 (l_s^T)/(l_o^M )-c_2=δ

where c_1 and c_2 are the coefficients of the line equation and δ is the deviation from the linear fit. This linear fit allowed for a parameter transformation in the optimal control problem to improve the numerical condition of the problem.

In Eq. (5), the first term, the error is not clearly defined.

Thank you for pointing this out. However, the e_error term was defined previously (see sentence below). Please let me know if this is not clear (lines 228-231):

Additional path constraints bounded: (i) the difference between the muscle-generated joint torques and the inverse dynamics torques (T_error), (ii) the difference between the modeled muscle excitations and the EMG signals scaled by k_EMG (e_error), [...].

In lines 211 and 212, what does it mean "modeled implicitly in the path constraints"? please explain "path" constraints and "implicitly" modelled.

Thank you for the comment. “Path” refers to the type of constraint applied to a trajectory/variable that the optimal control problem needs to follow. Since this is the main type of constraint used in the problem, we will omit “path” to avoid any confusions. Regarding “implicit modelling”, this concept is explained in detail by De Groote et al. (2016), which is actually a more relevant reference compared to the one that was previously cited in the manuscript (Van den Bogert et al., 2011). Thank you again for pointing this out, because this allowed us to cite the most relevant work about implicit modelling. Since we would prefer not to disrupt the flow of the explanation in this paragraph, we propose to change the text as follows (please see below – lines 224-228). We meticulously adopted the formulation described by De Groote et al. (2016), and therefore, we believe that directing the reader to this reference is the best way to communicate the concept, without disrupting the flow of the text.

The activation dynamics was implemented explicitly using (1), whereas the contraction dynamics was modeled implicitly in the path problem constraints, following the formulation developed by [42], to improve the efficiency and robustness of the problem. In fact, this simplified the calculation of the nonlinear equations describing muscles’ contractions.

In Eqs. (2,3), the variable Fm_max is not defined. In addition, is this variable a constant in this study? Why not include Fm_max as a model parameter? Between different subjects,

Fm_max is changing.

That is a very good point, thank you for raising it. Regarding the definition of Fm_max, it is defined in the second paragraph of the introduction. But please let me know if this is not clear. Regarding the inter-subject variability of Fm_max, you are absolutely correct. We explained in the discussion that we computed the maximum isometric forces of the muscles using the regression model described by Handsfield et al. (2014). However, we did not explain this in the methodology. Please find below the added explanation that elucidates this concept (lines 191-193):

where, F^M and F^T are the muscle and tendon forces, respectively. The F_max^M parameters for each muscle were computed using regression equations that relate muscle volume to each subject’s mass and height [39].

Reviewer #2: major comments

My primary concern regarding this manuscript is that the authors’ conclusions do not seem to fully align with the data presented. Initially, I was also confused about the graphs presented. This is partially because the image quality in the proofs makes the images difficult to read, but it took me a long time to realize that Fig 2 (mean joint torques) does not contain the same information as Fig 4 (joint angle time histories).

Thank you very much for pointing out the low quality of the images in the PDF. We believe the journal lowered the quality of the figures in the PDF to avoid an excessively high file size. If you click on the figure links on the top-right corners of the pages, you should be able to download the high-quality versions of the figures. Hopefully, that will work.

Clarity could be enhanced by adding titles to the images, since the unitless nRMSE values do not convey any information about the quantity being presented.

That is a very nice suggestion, thank you. As suggested, we added titles to the figures to address this issue.

Also, considering that the joint torques are a key outcome of this study, it would be helpful to include the time history plots for the joint torques (i.e., in the same format as Fig 4), at least as supplementary material.

Thank you for pointing this out. We included all time history plots for the joint torques in both the tracking simulations (Figs S3-S6) and predictive simulations (Figs S8-S11).

Aside from these misunderstandings, some of the conclusions in the paper do not seem to be well-supported. First, the paper only reports normalized (nRMSE) differences. This may obscure some substantial differences between the models. For example, in Fig 2a, the differences appear very large, but hip internal/external rotation torques are small in magnitude. Conversely, the differences in knee flexion/extension appear small, but may in fact be large in magnitude since knee flexion/extension torques are large during gait. It would enhance clarity to include non-normalized versions of Figs 2 and 3, perhaps as additional supplementary material.

Thank you for flagging this. We included in the supplementary material versions of Figures 2 and 3 that are normalized by body mass rather than the maximum experimental torques (Fig S2 and Fig S7, respectively). This will allow the reader to identify which joints have the largest torque RMSE in magnitude. Since we are including the RMSEs of multiple subjects, we divided the RMSE values by body mass.

Similarly, in the third paragraph of Results > Gait Predictive Simulations, the authors state that “the kinematics was predicted slightly more accurately by NSMs compared to FCM0 (Fig 4).” Looking at Fig 4, I would characterize the differences in knee flexion as dramatically different. Knee flexion is the largest magnitude joint angle in gait, and hthe experimental as the clearest pattern during gait. The results from FCM0 are missing the characteristic peak, differing from both the NSMs and experimental data by what looks to be 30-50 degrees. There are similarly large differences in other joints, including HipAA, Hip IE, and to some extent HipFE. This mischaracterization of the joint angles casts doubt on the authors’ interpretation of the joint torque results in Figs 2 and 3, leading the reader to wonder if similarly dramatic differences are somewhat hidden by the nRMSE calculations.

Thank you for pointing this out. According to the reviewer’s statement, we updated the text in the Results section as follows (lines 363-374):

Furthermore, the kinematics was predicted more accurately by NSMs, compared to FCM0 (Fig 4). In particular, the FCMs consistently under-flexed the knee during swing. In fact, in subjects 1 and 2, the kneeFE angles of FCM0 were missing the characteristic peak observed in both the experimental data and the simulations using NSMs. NSMs and FCM0 had difficulty in accurately predicting the hipAA angle for subjects 2 and 3, whereas, for subject 1, NSMs predicted it more accurately than FCM0. Additionally, both models consistently overestimated the ankle plantar flexion at the beginning of the gait cycle. In general, the most-accurately predicted angle by both models was the hipFE angle. Nevertheless, in subjects 1 and 2, the left hipFE profiles of the NSMs were closer to the experimental data, compared to FCM0. The predicted kinematics using FCM1, FCM2 and FCM3 are shown in the supplementary material (Fig S13-S15 in S1 File).

Another concern is related to the language used to describe the results. The authors provide a good overview of the literature in which similar studies used joint torques to “validate” the musculotendon parameter calibration. Nonetheless, I would argue that the only true gold standard for “validation” of the musculotendon parameters would be direct measurement from medical imaging to calculate the lengths of the muscle fibers and tendons. While the optimization procedure may produce more accurate joint torques, the muscle and tendon lengths may not be closer to the actual lengths in the human subject. I recommend the authors state that the results were “evaluated” by comparing to joint torques, rather than “validated” to avoid confusion.

That is a very valid concern, thank you for the comment. As suggested, we used the term “evaluation” instead of “validation” when comparing models in tracking simulations.

Related, there are a number of statements in the Discussion and Conclusion that results were “more accurate” or “less accurate” compared to some other quantity. The manuscript would be more clear if the authors stated precisely what was different, e.g., “correlation was significantly higher” or “nRMSE was significantly lower”. Please review the manuscript and address all occurrences of this issue.

Thank you for raising this issue. We made the suggested changes in the Discussion and Conclusion sections accordingly, especially when we were referring to specific results. Please find below the updated text:

(lines 387-389). In fact, in the DOFs where the predicted kinematics was considerably different from the experimental data, a higher nRMSE was found for the corresponding torques.

(lines 393-396). On the other hand, when the kinematics was predicted similarly by both models, there were either no significant differences between them in the corresponding torque nRMSE values (i.e., hipFE) or FCM-baseline yielded significantly lower nRMSE values (i.e., anklePD).

(lines 440-442). Differently from FCM-baseline, the muscles excitations generated by FCM-bounds in the gait predictions had significantly lower cross-correlation values compared to NSMs (Table 4).

(lines 456-458). Our results agree with their statement, as FCM-gait produced significantly lower nRMSE and higher R2 values for kneeFE torques compared to NSMs in the tracking simulations—a trend that was not observed with FCM-baseline.

(lines 476-478). In the predictive simulations, FCM-sagittal unexpectedly led to lower nRMSE and higher R2 values for hipAA torques, compared to FCM-baseline (Table 3).

(lines 555-557).Our results suggest that in the tracking simulations, the FCMs generally led to significantly lower nRMSE and higher R2 values between the experimental and estimated torques compared to the NSMs.

The first paragraph of the Introduction should include

---

## [Decision Letter · Decision Letter 1]

10 Jun 2025

Calibrated muscle models improve tracking simulations without enhancing gait predictions

PONE-D-24-57517R1

Dear Dr. Maceratesi,

We’re pleased to inform you that your manuscript has been judged scientifically suitable for publication and will be formally accepted for publication once it meets all outstanding technical requirements.

Kind regards,

Xianlian Zhou

Academic Editor

PLOS ONE

Additional Editor Comments (optional):

Reviewer two mentioned a few minor issues below, please correct these issues in the final manuscript.

I noticed a couple of remaining minor issues:

- Should the names of the tracking simulations TS0, TS1, etc be updated to match the new FCM-baseline, FCM-gait, etc?

- In S2 File, why are there NaN values for some joint torques during gait? Were these omitted from the analysis?

- There are some small typos such as double quotes facing the wrong direction in some of the edits made after revision

Reviewers' comments:

Reviewer's Responses to Questions

**Comments to the Author**

1. If the authors have adequately addressed your comments raised in a previous round of review and you feel that this manuscript is now acceptable for publication, you may indicate that here to bypass the “Comments to the Author” section, enter your conflict of interest statement in the “Confidential to Editor” section, and submit your "Accept" recommendation.

Reviewer #1: All comments have been addressed

Reviewer #2: All comments have been addressed

2. Is the manuscript technically sound, and do the data support the conclusions?

Reviewer #1: Yes

Reviewer #2: Yes

3. Has the statistical analysis been performed appropriately and rigorously? 

Reviewer #1: Yes

Reviewer #2: Yes

4. Have the authors made all data underlying the findings in their manuscript fully available?

Reviewer #1: Yes

Reviewer #2: Yes

5. Is the manuscript presented in an intelligible fashion and written in standard English?

Reviewer #1: Yes

Reviewer #2: Yes

6. Review Comments to the Author

Reviewer #1: Authors have successfully addressed my comments. All my concerns have been solved. It can be accepted now.

Reviewer #2: Thanks to the authors for their careful attention to the issues raised during review. The clarity of the manuscript has been much improved. I noticed a couple of remaining minor issues:

- Should the names of the tracking simulations TS0, TS1, etc be updated to match the new FCM-baseline, FCM-gait, etc?

- In S2 File, why are there NaN values for some joint torques during gait? Were these omitted from the analysis?

- There are some small typos such as double quotes facing the wrong direction in some of the edits made after revision

7. PLOS authors have the option to publish the peer review history of their article (what does this mean?). If published, this will include your full peer review and any attached files.

Reviewer #1: **Yes: **Yujiang Xiang

Reviewer #2: No

---

## [Editor Report · Acceptance letter]

PONE-D-24-57517R1

PLOS ONE

Dear Dr. Maceratesi,

I'm pleased to inform you that your manuscript has been deemed suitable for publication in PLOS ONE. Congratulations! Your manuscript is now being handed over to our production team.

Kind regards,

on behalf of

Dr. Xianlian Zhou

Academic Editor

PLOS ONE